# Interference of *Streptococcus agalactiae* Blitz Therapy in *Staphylococcus aureus* Microbiological Diagnosis in Subclinical Bovine Mastitis

**DOI:** 10.3390/vetsci11060233

**Published:** 2024-05-22

**Authors:** Ana Flávia Novaes Gomes, Fúlvia de Fátima Almeida de Castro, Márcio Roberto Silva, Carla Christine Lange, João Batista Ribeiro, Alessandro de Sá Guimarães, Guilherme Nunes de Souza

**Affiliations:** 1Faculty of Pharmacy, Universidade Federal de Juiz de Fora, Juiz de Fora 36038-330, Brazil; anaflavia.novaes@estudante.ufjf.br (A.F.N.G.); fulvia.almeida@estudante.ufjf.br (F.d.F.A.d.C.); marcio-roberto.silva@embrapa.br (M.R.S.); carla.lange@embrapa.br (C.C.L.); joao-batista.ribeiro@embrapa.br (J.B.R.); 2Brazilian Agricultural Research Corporation, Juiz de Fora 36038-330, Brazil; alessandro.guimaraes@embrapa.br

**Keywords:** mammary gland, contagious mastitis, diagnostic method, somatic cell count

## Abstract

**Simple Summary:**

In this study, we looked to evaluate the variation in sensitivity, specificity, predictive values, and accuracy (Kappa index) of the microbiological diagnosis for *Staphylococcus aureus* in a dairy cattle herd subjected to blitz therapy to eradicate *Streptococcus agalactiae*. The studied dairy herd had an average of 160 lactating Holstein cows and microbiological diagnosis was carried out on all lactating cows for 5 consecutive months. After treating the animals that presented a pure culture of *S. agalactiae* in microbiological diagnosis in the first and second milk sample collections, isolation of *S. aureus* was observed in these animals in the following milk sample collections, increasing the sensitivity of the microbiological diagnosis. The results indicate that due to the release of *S. agalactiae* being greater than that of *S. aureus*, due to competition in a nutritive culture medium, it was not possible to isolate *S. aureus* in the first collections. However, after the elimination of *S. agalactiae* through intramammary antimicrobial treatment, *S. aureus* began to be identified in the microbiological examination as there was no longer competition between the pathogens in the nutritive culture medium.

**Abstract:**

Bovine mastitis is an important and costly disease to dairy cattle. Diagnostic methods usually performed in Brazil are somatic cell counts (SCC) and milk microbiology. Low bacteria shedding in milk implies no colony growth in microbiological tests and false negative results. *Streptococcus agalactiae* and *Staphylococcus aureus* are principal pathogens of mixed mastitis. However, *S. agalactiae* has a higher bacterial release from the mammary gland than *S. aureus*, affecting microbiological sensitivity to diagnose *S. aureus*. This study aimed to estimate the SCC and total bacterial count (TBC) from cows according to pathogen isolated in milk and to evaluate variation in *S. aureus* diagnosis by a microbiological test during *S. agalactiae* treatment, which is called blitz therapy. Both *S. agalactiae* and *S. aureus* presented high SCC means, although *S. agalactiae* showed shedding of bacteria 2.3 times greater than *S. aureus*. Microbiological sensitivity to *S. aureus* increased for 5 months during *S. agalactiae* treatment. The prevalence of *S. agalactiae* fell after 5 months of therapeutic procedures. The prevalence of *S. aureus* increased to 39.0. The results showed that due to high sensitivity, the polymerase chain reaction (PCR) could be used at the beginning of blitz therapy with the goal of *S. agalactiae* eradication from the dairy herd.

## 1. Introduction

Bovine mastitis is an inflammation in mammary gland tissue and it is the most frequent and costly disease in dairy cows [1,2]. The main mastitis pathogens are *Staphylococcus aureus* and *Streptococcus agalactiae*. These bacteria usually cause persistent subclinical mastitis, that is, chronic intramammary infection [3,4]. Mastitis from these bacteria is usually endemic in developing countries such as Brazil [3,5].

Since *S. aureus* and *S. agalactiae* cause contagious mastitis, the main source of infection is infected cows and transmission occurs during milking [6,7]. Due to this, preventive measures for contagious pathogens are the adoption of the “ten-point plan”: adequate milking procedures; dry cow therapy; management of clinical mastitis in lactation; maintenance of milking equipment; biosecurity and disposal of chronically infected cows; goal setting; maintaining a clean and comfortable environment; mastitis records plan; monitoring mammary gland health indices; and periodic review of the control plan [8]. However, treatment for these two bacteria is distinct. *S. agalactiae* is an obligate intramammary pathogen and can be eradicated from a herd with blitz therapy: a protocol of intramammary antibiotics therapy in all mammary quarters of the infected animal [9,10]. On the other hand, infection from *S. aureus* is not very responsive to antibiotic therapy and the appropriate treatment conducted is anticipated dry therapy or culling cows with chronic cases [3,11]. In developed countries, preventive measurements for contagious pathogens as well as *S. agalactiae* eradication are performed widely [3].

Proper diagnosis of these bacteria is necessary to implement treatment and preventive procedures. The most used diagnostic methods to identify mammary gland inflammation and the responsible pathogen for intramammary infection are somatic cell count (SCC) in milk and microbiological exams, respectively [12]. Mastitis pathogens present different tropisms to the mammary tissue, developing variation in inflammatory response levels, and consequently, the range of SCC mean per pathogen [13,14,15]. Furthermore, bacteria are shed in different patterns by the mammary quarter from naturally infected cows [16]. It has been reported that *S. agalactiae* is shed by mammary glands in higher quantities than *S. aureus* [16], which might impair the culturing of *S. aureus* in microbiological tests. Moreover, *S. aureus* has a particular intermittent shedding by mammary glands with phases of low or high bacteria release in milk [3,7]. Low bacterial or intermittent shedding may be due to not having the required number of bacteria in the milk sample to promote bacterial growth in microbiological exams, interfering with the sensitivity of the test [17]. In that case, a microbiological exam could present false negative results [2]. Multiple milk sample collection on different days is a strategy to increase microbiological exam sensitivity to *S. aureus* during the intermittent shedding cycle [17,18]. However, in cows with mixed infections (coinfection), the high quantity of *S. agalactiae* in milk samples still could be a cause of the low sensitivity of *S. aureus* culture.

In this context, this study aimed to estimate the SCC means and bacterial shedding patterns according to pathogens isolated in composite milk samples. In addition, we also evaluate the sensitivity, specificity, predictive values, and accuracy (Kappa index) of microbiological diagnosis of *S. aureus* during blitz therapy to eradicate *S. agalactiae* considering cows with mixed infections.

## 2. Material and Methods

### 2.1. Ethical Committee for Animal Use

This study was approved by the Ethical Committee for Animal Use of Embrapa Gado de Leite, protocol number 8308220322. This study was carried out in a commercial dairy herd with the owner’s agreement.

### 2.2. Characteristics of Studied Dairy Herd

One dairy herd was selected based on a high SCC in bulk milk tank samples and *S. aureus* and *S. agalactiae* microbiological isolation in selective media (salt mannitol and TKT agar). The herd selected was located in Minas Gerais state, Brazil. Animals were Holstein breed, housed in a Free Stall system and submitted to three mechanic milkings per day. Nutritional management was based on maize silage, concentrate, and mineral salt. Official milk control was performed monthly. The monthly average of milked cows was 160 with an average lactation of 305 days and production of 8.050 kg.

### 2.3. Frequency and Procedures for Collecting Milk Samples

During the study period, 737 milk samples were collected over 5 months for subclinical mastitis microbiological diagnosis. However, animals with clinical mastitis or under treatment with antibiotics were excluded. From each milking cow, a composite milk sample was obtained for microbiological exam, SCC, and total bacterial count (TBC). Milk for microbiological examination was collected in duplicate in a sterile recipient, after teat antisepsis with 70% alcohol. Milk for SCC and TBC analysis was taken down from the milking machine collector receptacle, in recipients with Bronopol^®^ and Azidiol^®^ preservatives, respectively. After the procedures, the obtained samples were transported in isothermal boxes with recycled ice to Embrapa Dairy Cattle, where analyses were performed. After 21 to 30 days, a new microbiological collection procedure was performed in all lactating cows. The protocols of milk collection, microbiological exam, and blitz therapy were conducted monthly for five months. During this study, preventive measures and appropriate milking procedures were established to avoid new intramammary infections.

### 2.4. Methodology for Somatic Cell Counts (SCC), Total Bacteria Counts (TBC), and Microbiological Diagnosis

The method used to establish the SCC and TBC was flow cytometry (Bentley^®^). Equipment for TBC analyses was calibrated for the bacterial cell direct count. Microbiological milk examination occurred according to recommendations by the National Mastitis Council protocol [19], which combines the use of nutrient media with biochemical tests for the detection of different mastitis pathogens. At first, each sample was inoculated in 10 µL of the blood agar medium. After inoculation, the plates were incubated in an oven at 37 °C. The first evaluation was taken after 24 h of incubation and the second evaluation after 48 h. It was assessed whether there were growth and colony characteristics. After identifying the colonies, they were picked up on plates with BHI agar medium (Brain Heart Infusion), followed by a new incubation at 37 °C. This pick-up in a new medium was carried out to obtain a sufficient number of colonies to carry out the other microbiological and biochemical tests. Colonies isolated on BHI agar were subjected to Gram staining and catalase testing. Gram-positive and catalase-positive samples were requested for coagulase testing and Voges Proskauer to identify *Staphylococcus aureus*. While Gram-positive and catalase-negative samples were caused by CAMP tests, hydrolysis of sodium hippurate and esculin was used to identify *Streptococcus agalactiae*. Individual milk samples with the isolation of 3 or more different colonies were determined to be contaminated. The SCC and TBC average was estimated according to single isolated pathogens.

### 2.5. Treatment of Subclinical Mastitis Cases

Considering the microbiological results, all animals that had positive samples of *S. agalactiae* were submitted to blitz therapy. Treatment was performed by intramammary administrations of 75 mg ampicillin and 200 mg cloxacillin (Bovigan, Bayer^®^) in all mammary quarters. A total of 3 antibiotic administrations were carried out in 36 h. According to the withdrawal period recommended by the drug fabricant, the milk withdrawal time was 4.5 days.

### 2.6. Study Design and Statistical Analysis

The study design consisted of prevalence studies between lactating cows to subclinical mastitis caused by *S. aureus* and *S. agalactiae* according to months (5 months) in a dairy herd to evaluate over time the microbiological diagnosis of *S. aureus* during blitz therapy to eradicate *S. agalactiae* considering cows with mixed infections. In statistical analyses, the first step was to evaluate the normality of the distribution of SCC and TBC data using the Shapiro–Wilk test. If the SCC and TBC data did not present normal distribution, these results were transformed to Log 10 basis and the Shapiro–Wilk test was applied again. After transformation to Log 10, if the results presented a normal distribution, Analysis of Variance (ANOVA) was applied, and a post hoc Student *t*-test was conducted on independent samples to compare means. However, if after transformation to the Log 10 basis, the results did not present a normal distribution, a Kruskal–Wallis test was applied for independent multiple samples, and a post hoc Mann–Whitney test was carried out to compare medians. After comparisons of transformed SCC and TBC data, the means and medians were back-transformed for the presentation of the results. The means and medians of the SCC and TBC, respectively, were compared according to subclinical mastitis pathogens only in samples with one type of colony growth.

Real prevalence (RP) of *S. aureus* and *S. agalactiae* according to months during blitz therapy were estimated according to Habibzadeh et al. (2022) [20] using the apparent prevalence (AP), sensitivity (SEN), and specificity (SPC) found in this study for *S. aureus* and the results obtained by Keefe (1997) [21] to *S. agalactiae*. The calculations are presented below.


AP = number of positive cows by microbiological diagnosis/total number of cows RP = (AP + SPC − 1)/(SEN + SPC − 1)


The dynamic of subclinical mastitis infections for each cow during each month of the study was represented by a comparison of 2 consecutive months, evaluating the microbiological diagnosis of the current month in relation to the previous month and considering the presence and absence of *S. aureus* or *S. agalactiae*. This classification was carried out as follows: (1) incidence of subclinical mastitis, absence of *S. aureus* or *S. agalactiae* in the previous month but presence in the following month; (2) elimination of subclinical mastitis, presence in the previous month but absence in the following month; and (3) chronic subclinical infection, presence in both of the 2 consecutive sample months. Due to the number of lactating cows to have changed over time, the incidence of subclinical mastitis, elimination of subclinical mastitis, and chronic subclinical infection were estimated per 100 cows per month to compare according to months.

During the 5 months of study, the microbiological results for *S. aureus* were compared considering the previous and next month (Figure 1 and Table 1) to estimate sensitivity, specificity, positive predictive value, negative predictive value, accuracy, and a confidence interval of 95%.

The confidence intervals (95%) of sensitivity, specificity, positive predictive value, negative predictive value, and accuracy (Kappa index) were estimated (CI 95% = mean ± 1.96*standard deviation) for the first 4 months of the study. Statistical analysis was performed in SPSS 22 (IBM^®^).

## 3. Results

### 3.1. Somatic Cell Counts and Total Bacteria Count according to Subclinical Mastitis Pathogens

Measurement of the SCC and TBC represented inflammatory cells and bacterial shedding quantity in milk samples, respectively. However, the SCC presented a normal distribution after transformation to a Log 10 basis but the TBC did not present a normal distribution. To estimate the SCC means and TBC medians per pathogen, only samples with pure growth of one colony were considered. *S. aureus*, *S. agalactiae*, *Streptococcus uberis*, coagulase-negative staphylococci, and *Corynebacterium bovis* presented pure culture. The SCC means (Table 2) for *S. agalactiae*, *S. aureus,* and *S. uberis* presented high SCC means and were statistically equal (*p* > 0.05). These pathogens showed statistical differences (*p* < 0.05) of minor pathogens and no growth samples, which developed low SCC means. Bacterial release from the mammary gland was evaluated through TBC medians results. The greater bacterial shedding in milk was represented by *S. agalactiae* and *S. uberis* (Table 2). While the lowest TBC medians were obtained in “no growth samples”, *C. bovis* and coagulase-negative staphylococci. *S. aureus* presented a TBC median statistically equal to *Streptococcus* spp.

### 3.2. Evaluation of the Dynamic of Subclinical Mastitis Infection according to Pathogens

In a comparison of the first and second microbiological cultures, 19/89 viable milk samples that presented pure growth of *S. agalactiae* in the first month showed isolation of *S. aureus* after the first blitz therapy procedure (Table 3). Of the 19 milk samples, 15 had pure isolation of *S. aureus* and 4 had isolation of *S. aureus* and *S. agalactiae*. At the same time, the incidence of *S. aureus* increased by 19.4%: 28 of 144 lactating cows did not present *S. aureus* in the first evaluation but showed growth of this bacterium on the second microbiological exam. In contrast, the rate of *S. agalactiae* elimination was 40.3%: therapy was efficient in 58 of 144 cows. Comparisons of the following months had a lower incidence of *S. aureus* and a decline in the elimination of *S. agalactiae* rates. *S. aureus* incidence in the following comparisons was 3.1%, 10.3%, and 8.9%. The elimination rate of *S. agalactiae* was 7.6%, 2.1%, and 1.9%. The rate of cows chronically infected by *S. aureus* increased from 19.4% to 28.7% at the end of this study.

Through the results of sensitivity and specificity, the real prevalence of *S. aureus* was estimated. The real prevalence of *S. agalactiae* was calculated according to the sensitivity and specificity reported by Keefe (1997). In the first result, *S. agalactiae* presented a real prevalence of 61%, while the prevalence of *S. aureus* was 25.2% (Table 3). After blitz therapy, the real prevalence of *S. agalactiae* was 1.4%, and was 39.0% for *S. aureus*. During *S. agalactiae* mastitis treatment, the prevalence of this pathogen dropped to 59.6%. In contrast, the real prevalence of *S. aureus* increased by 13.8% (Table 3).

### 3.3. Variation of Sensitivity, Specificity, Predictive Values, and Accuracy of Microbiological Diagnosis of S. aureus during Blitz Therapy

Milk microbiological procedures were performed monthly for five months and results were compared with data from the month before. Four comparisons of successive months were made. Data from the microbiological exam was used to evaluate the test to diagnose *S. aureus* (Table 4). A comparison of the first and second months showed a sensitivity of 50.0% and it increased to 89.7% in the next comparison after the first procedure of *S. agalactiae* treatment. The sensitivity found after 5 months was 76.3%. The specificity increased from 79.0% to 91.4% in the first and last month’s comparison. Positive and negative predictive values first presented as 62.2% and 69.5% and then were enhanced to 85.7% and 85.2%, respectively, by around the last month. The accuracy of the microbiological exam revealed an increase of 18.3% during the 5 months of blitz therapy, ranging from 67.1% to 85.4% in the first and last comparisons.

## 4. Discussion

The main sources of contagious mastitis pathogens *S. agalactiae* and *S. aureus* are infected cows and mixed intramammary infection by these bacteria. Moreover, subclinical mastitis by both pathogens can lead to high levels of SCC [3]. In this study, *S. agalactiae*, *S. uberis,* and *S. aureus* presented the highest SCC means, agreeing with previous studies performed by Souza et al. (2009) and Souza et al. (2016) [2,13]. *S. agalactiae* and *S. uberis* also showed greater TBC median results, which means that these bacteria were shed in higher quantities in milk, as related before by Lopes Jr. et al. (2012) [16]. Not surprisingly, the *S. aureus* TBC median was statistically equal to pathogens with a greater TBC and with no growth samples, with a low TBC median. These findings demonstrated the already related *S. aureus* dynamic of intermittent shedding by the mammary gland [7]. Nevertheless, the shedding of *S. agalactiae* found in this study was 2.3 times greater than *S. aureus*. This was not unexpected since in Lopes Jr. et al.’s (2012) [16] study, *S. agalactiae* shedding was 4.3 times higher than *S. aureus* in milk from infected mammary quarters. These results show that in mixed intramammary infections by both pathogens, *S. agalactiae* might have more probability of growth than *S. aureus* when milk is cultured in nutritive media. As regards microbiological exams, it is known that a low quantity of bacteria in samples impairs pathogen isolation and has a negative effect on test sensitivity [12,17]. In that case, besides *S. aureus* having intermittent shedding as the cause of low sensitivity in culture, *S. agalactiae* might also play an important role in the interference of *S. aureus* diagnosis. The high quantity of *S. agalactiae* in the milk sample could mask *S. aureus* isolation, resulting in *S. agalactiae* isolation only.

The sensitivity of the microbiological test to *S. aureus* was influenced by the presence of *S. agalactiae* as part of the mixed intramammary infection. At the beginning of the study, the sensitivity to *S. aureus* diagnosis was low (50.0%) but suddenly increased to 89.7% after only one *S. agalactiae* therapy procedure. This initial rise occurred at the same time as the peak of *S. agalactiae* elimination (40.3%). However, in the last comparison of the study, after 5 therapy protocols, the sensitivity to *S. aureus* stagnated at 76.3%, a reasonable rate when compared to the first finding. The specificity of *S. aureus* isolation increased as well during blitz therapy, and at the finish of the study, reached a satisfactory result of 91.4%. The predictive values demonstrate the rate of truly positive or negative results in the field. These parameters also raised consistent results of 85.7% and 85.2% to positive and negative predictive results. Finally, the accuracy of the microbiological test for *S. aureus* had a significant improvement of 13.8% during the study and showed a final result of 85.4%. All test parameters analyzed showed better outcomes at the end of the blitz therapy, reflecting an upgrade of *S. aureus* microbiological diagnosis after *S. agalactiae* treatment.

The interference of *S. agalactiae* in *S. aureus* diagnosis was proved when 19 of 89 (21.3%) viable milk samples yielded only *S. agalactiae* before blitz therapy but were positive for *S. aureus* in the second month, after 1 treatment procedure. The *S. aureus* incidence rate was affected since 28 of 144 (19.4%) cows were diagnosed with intramammary infection by this pathogen in the second month. Of these 28 cows, 19 were first diagnosed with *S. agalactiae*, and the other 9 of them were probably misdiagnosed in the first month, likely due to intermittent cycle of *S. aureus*. At the same time, 58 of 144 cows (40.3%) had a good response to blitz therapy because the second microbiological culture did not show *S. agalactiae* as the first culture did. In the course of therapy protocols, the *S. aureus* incidence was lower than the first comparison but still significant. That might have occurred due to progressive elimination, and in consequence, low prevalence of *S. agalactiae*. During this study, rigorous preventive measures for contagious mastitis were applied in the studied herd. For this reason, fluctuations in *S. aureus* incidence might be assigned to *S. agalactiae* elimination and misdiagnosed by intermittent shedding and, probably, it was not a real incidence. Furthermore, the rate of cows chronically infected by *S. aureus* also increased in the course of this study. This was expected since persistent infections are typical of *S. aureus* and a factor of interference in microbiological diagnosis was removed.

Real prevalence was estimated for each pathogen considering the microbiological parameters results, such as sensitivity and specificity. While apparent prevalence only considered the percentage of infected animals in the herd, the real prevalence also takes into account the diagnostic method utilized. For *S. agalactiae*, the results of the apparent and real prevalence showed slight differences among them, but both declined with blitz therapy as expected. Meanwhile, the apparent and real prevalence of *S. aureus* increased during the study. In contrast to *S. agalactiae*, both *S. aureus* prevalence had significant divergence at some point. In the second month, the *S. aureus* apparent prevalence was 38.9% while the real prevalence was 25.5%. Another difference that must be highlighted was the increase of 10.6% in the apparent prevalence from the first month to the second, when the real prevalence rose only by 0.3%. That could be explained since some animals were infected by *S. aureus* in the first procedure, but the pathogen was not diagnosed. Real prevalence considers gaps in microbiological tests, and due to this, the real prevalence between the first and second months did not have a difference. In the other months, the real prevalence of *S. aureus* was higher than the apparent prevalence, presumably due to the interference of the intermittent shedding pattern.

In developing countries, a microbiological exam is the most used method to identify pathogens responsible for mastitis. Despite being a more expensive method, PCR could be used to diagnose *S. aureus* even in phases of low bacterial shedding or in mixed intramammary infections with *S. agalactiae*, complementary to a traditional microbiological test, at the beginning of the blitz therapy to eradicate *S. agalactiae* from the dairy herd.

## 5. Conclusions

In summary, this study showed that *S. agalactiae* was shed by mammary glands in higher quantities than *S. aureus*. As a consequence, eradication of *S. agalactiae* implied an improvement in the probability of isolating *S. aureus* and increased sensitivity, specificity, predictive values, and accuracy of microbiological tests to diagnose *S. aureus* from bovine mastitis. Since microbiological tests had implications for the sensitivity of *S. aureus*, PCR could be used complementary to traditional microbiological testing to identify both contagious pathogens at the beginning of the blitz therapy.

## Figures and Tables

**Figure 1 vetsci-11-00233-f001:**
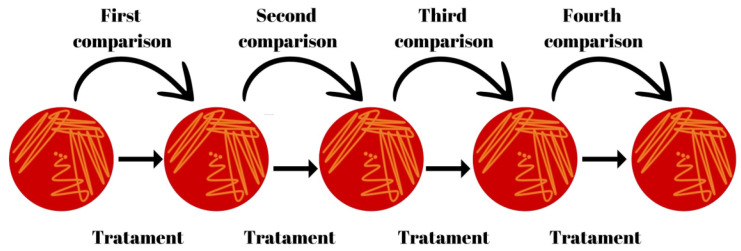
Collection scheme and comparison for each lactating cow among months of the study.

**Table 1 vetsci-11-00233-t001:** Contingency table for evaluation of sensitivity, specificity, positive predictive value, negative predictive value, and accuracy (Kappa index) for *S. aureus* were compared considering the previous and next month.

Microbiological Results of the Previous Month	Microbiological Results for Next Month	
	Positive	Negative
Positive	a	b
Negative	c	d

Where N = a + b + c + d; sensitivity: a/(a + c); specificity: d/(b + d); positive predictive value (PPV): a/(a + b); negative predictive value (NPV): d/(c + d); accuracy (Kappa index): (a + d)/N.

**Table 2 vetsci-11-00233-t002:** Results of somatic cell count (SCC) means and total bacterial count (TBC) medians by isolated pathogens in the microbiological exam of composite milk samples.

Microbiological Result	N	Mean of SCC (×1000 Cells/mL) ^1^	Median of TBC (×1000 cfu/mL) ^2^
No growth	21	139 ^a^	27 ^a^
*Corynebacterium bovis*	14	134 ^a^	27 ^a^
Coagulase-negative staphylococci	12	151 ^a^	27 ^a^
*Staphylococcus aureus*	11	730 ^b^	41 ^ab^
*Streptococcus uberis*	12	1.318 ^b^	100 ^b^
*Streptococcus agalactiae*	50	1.658 ^b^	93 ^b^

N—number of milk samples; cfu—colony forming units. ^1^ Different letters according to mean of SCC results means statistical difference (*p* < 0.05) to ANOVA and Student’s *t*-test to independent samples. ^2^ Different letters according to median of TBC results means statistical difference (*p* < 0.05) to Kruskal–Wallis and Mann–Whitney test.

**Table 3 vetsci-11-00233-t003:** Variations of apparent prevalence (AP) and real prevalence (RP) of *Streptococcus agalactiae* and *Staphylococcus aureus* per month during blitz therapy. Incidence and elimination rate of *S. agalactiae* and incidence and chronic infected (CI) rate of *S. aureus* (100 cows per month).

Month	*S. agalactiae*		*S. aureus*		*S. agalactiae*		*S. aureus*	
	AP (%)	RP (%)	AP (%)	RP (%)	Incidence	Elimination	Incidence	CI
1	61.0	61.6	28.3	25.2	5.6	40.3	19.4	19.4
2	18.8	17.1	38.9	25.5	1.5	7.6	3.1	26.7
3	3.8	1.4	29.8	38.5	2.7	2.1	10.3	19.9
4	4.8	2.4	33.6	36.9	1.3	1.9	8.9	28.7
5	3.8	1.4	35.0	39.0	-	-	-	-

**Table 4 vetsci-11-00233-t004:** Variation of sensitivity, specificity, predictive values, and accuracy of microbiological diagnosis of *S. aureus* during blitz therapy.

Months	Sensitivity (%)	Specificity (%)	PPV (%)	NPV (%)	Accuracy (%)
					(Kappa Index)
1	50.0	79.0	62.2	69.5	67.1
2	89.7	78.5	62.5	95.0	81.7
3	59.1	88.6	74.3	79.5	78.1
4	76.3	91.4	85.7	85.2	85.4
Mean	68.8	84.4	71.2	82.3	78.1
SD	17.7	6.6	11.2	10.7	7.9
MCI 95%	34.1–100.0	71.4–97.3	49.2–93.1	61.4–100.0	62.6–93.6

SD—standard deviation; MCI 95%—confidence interval of 95%; PPV—positive predictive value; NPV—negative predictive value.

## Data Availability

The database used and analyzed during the current study is available from the corresponding author upon reasonable request.

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
