# Peer review of "Interference of Streptococcus agalactiae Blitz Therapy in Staphylococcus aureus Microbiological Diagnosis in Subclinical Bovine Mastitis"

_vetsci, 2024, doi:10.3390/vetsci11060233_

Round 1

Reviewer 1 Report

Comments and Suggestions for Authors

Dear Authors, 

the manuscript entitled  "Interference of Streptococcus agalactiae Blitz Therapy in Staphylococcus aureus Microbiological Diagnosis in Subclinical Bovine mastitis" is really interesting and deals with the most common disease responsible for  relevant econimic losses in dairy industries. In particular, the manuscript focuses on Straptococcus agalactiae and Staphylococcus aureus, the two causative pathogens of  mixed contagious mastitis. Obviously, the importance of a proper isolation and identification of these pathogens and a rapid diagnosis is necessary to guarantee an effective treatment and to prevent new cases of mastitis.

The manuscript is  well written. However, it needs more scientific rigor, in the description and presentation of both  Matherials and Methods and Rseults sections. In particular, to better describe these two  above-mentioned sections, it is desiderable to divide them into subsections. Otherwise, it becomes really difficult to the readers to really follow what is reported in the manuscript. 
In Matherials and Methods section, a description of the microbiological examination performed should be reported.  It is not enough to report only that  "Microbiological milk examination occurred according to recommendations by National Mastitis Council protocol (NMC, 2017)". So in this section  perfomed examination techinques should be better reported. Furthermore, it would be better not to reports results in this section. 

Two minor observations:

- make sure that the scientific names of bacterial species are written in italics in   in the whole manuscript.

- the surname  of first authors of the cited references should not be written with capital letters in main body of the manuscritpt. Please correct them. 

-remove actual before prevalence (lines 19 and 21).

I sincerely hope that these suggestions will enhance this manuscript. 

Comments on the Quality of English Language

The English language is fine. Only minor editing of the language is required. 

Author Response

Dear Reviewer 1,
Attached are the changes made to the manuscript.

Yours sincerely.

Guilherme Nunes de Souza

Reviewer 2 Report

Comments and Suggestions for Authors

REVIEW for the journal Veterinary Sciences (ISSN 2306-7381)

Article “Interference of Streptococcus agalactiae Blitz Therapy in Staphylococcus aureus Microbiological Diagnosis in Subclinical Bovine mastitis

Manuscript ID: vetsci-2957614

Authors:  Ana Flávia Novaes Gomes, Fúlvia de Fátima Almeida de Castro, Márcio Roberto Silva, Carla Christine Lange, João Batista Ribeiro, Alessandro de Sá Guimarães, Guilherme Nunes de Souza

            Brief summary. Bovine mastitis stands as one of the most prevalent and economically significant diseases affecting dairy cows. In cases of mixed infections, where multiple pathogens are involved, the presence of a high quantity of S. agalactiae in milk samples might still lead to a reduced sensitivity of S. aureus culture. The objective of the current study was to assess the mean somatic cell counts and bacterial shedding patterns based on the isolated pathogens in composite milk samples. Additionally, the authors examined the efficacy of microbiological testing in diagnosing S. aureus during rapid therapy aimed at eradicating S. agalactiae.

General concept comments

1.       Introduction. The introduction section of the article includes a comprehensive review of the relevant literature sources pertaining to the subject under analysis. Additionally, the article outlines its objectives, thereby setting the direction for the research.

The aim of this study was to estimate SCC means and bacterial shedding pattern according to pathogen isolated in composite milk samples. In addition, the authors  evaluated the microbiological test to diagnose S. aureus during blitz therapy to eradication of  S.agalactiae.

2.       Materials and Methods. One dairy herd (of Holstein  breed)  located in Minas Gerais state, Brazil was selected based on high SCC in bulk milk tank sample and on S. aureus and S. agalactiae microbiological isolation in selective media (salt mannitol and TKT  agar). Milk sampling was performed in all lactating cows for five months. From each milking cow, a composite milk sample was obtained for microbiological exam, SCC and  total bacterial count (TBC). Based on the microbiological results, all animals with positive samples for S. agalactiae underwent blitz therapy. This treatment involved intramammary administration of ampicillin (75 mg) and cloxacillin (200 mg) in all mammary quarters. Following a 21-day interval, a subsequent microbiological sampling procedure was conducted on all lactating cows. This protocol of milk collection, microbiological examination, and blitz therapy was repeated monthly for a duration of five months.

3.       Data management and statistical evaluation.  In my view, the authors have demonstrated a thoughtful selection of analytical tools that align with the specific goals outlined in their research. This enhances the overall quality and credibility of the study.

4.       From my perspective, the assessment of the results presented in the article aligns with the research objectives and methodology.

5.       The study's conclusions are in harmony with its predefined objectives. In summary, the authors indicate that S. agalactiae was shed in higher quantities from the mammary gland compared to S. aureus. Consequently, the eradication of S. agalactiae led to an enhanced probability of isolating S. aureus and improved sensitivity, specificity, predictive values, and accuracy of microbiological tests for diagnosing S. aureus in bovine mastitis. Given the implications of microbiological tests on the sensitivity of S. aureus, PCR could be used in conjunction with traditional microbiological testing to identify both contagious pathogens at the outset of blitz therapy.

Specific comments

1.    I overlooked the clear formulation of the research hypothesis.The methodology section necessitates a more comprehensive and organized approach. Furthermore, a detailed description of the study design is essential.

2.    In my view, the statistical analysis section could benefit from more detail, including explanations of all the statistical analysis methods employed and the corresponding metrics. The sample size should also be specified in the methodology.

3.    Lines 104-105: “The SCC results presented normal distribution”

Given the uncertainties, it was questioned whether presenting the SCC log10 data demonstrated a normal distribution and whether parametric analysis methods were suitable for this purpose. Common practice in this case is the transformation of SCC into SCS = (log2 (SCC/100)) + 3 [Ali A.K.A., Shook G.E. An optimum transformation for somatic cell concentration in milk. J. Dairy Sci. 1980; 63:487–490. doi: 10.3168/jds.S0022-0302(80)82959-6].

Conclusion.  The findings of this study indicated that polymerase chain reaction (PCR) assays can be used in the early stages of rapid treatment to eradicate S. agalactiae from dairy herds due to their increased sensitivity. The implications of these findings go beyond academia, resonating with stakeholders involved in dairy farming and those invested in promoting sustainable and premium milk production.

Sincerely, reviewer.

Author Response

Dear Reviewer 2,
Attached are the changes made to the manuscript.

Yours sincerely.

Guilherme Nunes de Souza

Reviewer 3 Report

Comments and Suggestions for Authors

This study estimated the SCC and total bacterial count (TBC) of dairy cows from pathogens isolated from cow's milk and evaluated changes in the diagnosis of Staphylococcus aureus by microbiological testing during treatment with Streptococcus agalactiae. However, there are still some issues that need to be determined.

1. Please indicate the number of experimental animals in the Material and Methods.

2. Line 111: "S. aureus" in italics please.

3. Line 167: " S. agalactie" in italics please.

4. Please keep the header of the table concise and put some elaboration in the table notes.

5. Does S. agalactie also contribute to lower S. aureus populations and thus low sensitivity to microbial testing by inhibiting the growth of S. aureus, rather than just lowering the rate of S. aureus shedding?

6. What is the mechanism of action of S. agalactie in reducing the rate of shedding of S. aureus? Do the authors have any more of the following studies? Please add additional experiments and discuss the previous studies in your discussion.

7. The article mentions the detection of S. aureus by PCR, but I did not see this method described in the material methods.

8. Reference 2, 14 missing page numbers.

Author Response

Dear Reviewer 3,
Attached are the changes made to the manuscript.

Yours sincerely.

Guilherme Nunes de Souza

Round 2

Reviewer 1 Report

Comments and Suggestions for Authors

Dear Authors, 

thank you for addressing all my requests. 

Comments on the Quality of English Language

A Minor English revision of the manuscript is requested, before publication.